# Consistency between inter-institutional panels using a three-level Angoff-standard setting in licensure tests of foreign-trained dentists in Sweden: A cohort study

Jesper Dalum[1]*, Liselotte Paulsson[2], Nikolaos Christidis[1], Mikael Andersson Franko[3], Klas Karlgren[4,5], Charlotte Leanderson[6], Gunilla Sandborgh-Englund[1]

1 Department of Dental Medicine, Division of Oral Diagnostics and Rehabilitation, Karolinska Institutet, Stockholm, Sweden, 2 Department of Orthodontics, Faculty of Odontology, Malmö University, Malmö, Sweden, 3 Department of Clinical Science and Education, Södersjukhuset, Karolinska Institutet, Stockholm, Sweden, 4 Department Learning, Informatics, Management and Ethics, Karolinska Institutet, Stockholm, Sweden, 5 Faculty of Health and Social Sciences, Western Norway University of Applied Sciences, Bergen, Norway, 6 Department of Neurobiology, Care Sciences and Society, Karolinska Institutet, Stockholm, Sweden

* jesper.dalum@ki.se

## Abstract

Licensure exams play a crucial role in ensuring the competence of individuals entering a profession, thereby safeguarding the public and maintaining the quality and integrity of the profession. In Sweden, dentists educated outside the European Union seeking to practise dentistry must undergo a re-certification process. The re-certification process includes a theoretical examination where pass marks are set using a three-level Angoff method. This study aimed to determine the consistency of the Angoff ratings using independent panels at two Swedish universities. Two cohorts of panellists were included in the study: one reference and one external. The reference panel was responsible for rating the upcoming theoretical examinations in the proficiency test, which were used to set the pass mark. The external panel, recruited from a dental school at a university in another region in Sweden, provided ratings after the examinations. Three examinations during 2019–2020 were included in this study (267 items in total). There was a strong correlation ($\rho \geq 0.70$, p < .001) between the ratings of the two independent panels, with no significant differences in item ratings across the full exams, dental disciplines, and professional qualifications analysed. This suggests that the three-level Angoff method reliably produces similar standards for assessing the competence of the minimally qualified dentist across different institutions. The expectations of the minimally qualified but still acceptable dentist were comparable between the two independent panels across the three theoretical examinations explored. The alignment between the panels indicates valid, reliable standards across institutions, despite the independent syllabi of the two study programmes. However, while there is an alignment, differences in ratings remain. Consequently, involving multiple institutions in future standard-setting processes could help ensure that the standards reflect a broader range of educational practices, supporting the credibility of licensure examinations.

**Data Availability Statement:** All data files are available from zenodo.org DOI https://doi.org/10.5281/zenodo.13951858.

**Funding:** The author(s) received no specific funding for this work.

**Competing interests:** The authors have declared that no competing interests exist.

## Introduction

Licensure has a crucial function in ensuring the competence of individuals entering a profession and safeguarding the public, consequently maintaining quality and integrity within the profession. Dentists educated outside of the European Union who seek to practise dentistry in Sweden can undergo a re-certification process for licensure [1, 2], which involves two examinations: a theoretical exam and a clinical skills assessment. The examinations are administered by the Department of Dental Medicine at Karolinska Institutet. After passing these examinations, a six-month clinical training period follows [1, 2]. The examinations used in the re-certification process are competency-based, and the blueprint and standard setting are aligned with the standards of a dentist in Sweden for licensure, i.e., the learning outcomes of a Swedish university degree in dental surgery [2, 3].

In competency-based proficiency tests, it is important to establish reliable and valid measures aligned with the expected standards [4, 5]. There are several methods to address the challenge of setting a defensible pass-mark based on the difficulty level of the test items [4], such as the Angoff [6], Nedelsky [7], and Ebel [8] methods. The Angoff method is commonly used to set the pass markon theoretical examinations [9–12]. In Angoff, a panel of subject matter experts are used to rate each individual question or task. The panel ratings represent what is expected of the minimally qualified but still acceptable participant [12] and is used to distinguish those who have the knowledge or skills required [13, 14]. The original Angoff method does come with challenges, such as the need for several subject matter experts, the difficulty for them to define the minimally qualified participant [15], and the difficulty to estimate how many of these participants would answer a specific task or question correctly [16]. To simplify the method, two major modifications have emerged: a two-level method, where panellists decide whether the minimally qualified but still acceptable individual would answer a test item correctly 'YES' or 'NO' [17], and a three-level version introducing a 'MAYBE' option [18, 19]. In the theoretical examination included in the proficiency tests of non-EU dentists, a simplified version of the three-level Angoff method is used for setting the pass mark. Training and consensus have been redacted, which serves to balance a manageable workload.

The Angoff-panel for the theoretical examination is currently recruited from a single university and concerns how the credibility and reliability of the pass mark on these nationwide examinations may arise. Whilst all Swedish dental programmes adhere to the intended learning outcomes of the Swedish Degree of Master of Science in Dental Surgery, which is required for a Swedish degree in dental surgery [3], they design their programme syllabi to achieve these outcomes independently. Thus, the question arises if the re-certification Ang off ratings and expectations of the minimally qualified are consistently aligned across all Swedish dentistry programmes, something this study aimed to determine as reflected by the ratings from two universities using the three-level Angoff method. The null hypothesis was that there would be no significant difference in the consistency between the two independent panels.

## Materials and methods

Two cohorts of panellists were included in the study, a reference and an external panel. The reference panel was used to consecutively rate the regular theoretical examinations of the proficiency test ratings used to set the pass mark of the exam. As part of the study, after the examinations, an external panel was recruited from a dental school at a university in another region in Sweden. This panel rated the same examinations as the reference panel. Both panels rated each question in each examination using a three-level Angoff method. Three examinations from the period 2019–2020 were included in this study (total n = 267 items).

The consistency of the item ratings was analysed for each full test, all items in total, and those items aligned to each dental discipline and learning outcome in the examinations. This was used to assess the method's reasonableness and explore its credibility at the item level [13, 20]. Additionally, the correlation between item difficulty and the Angoff panellists' mean item rating was analysed to explore whether participants reached the set Angoff standard within each dental discipline. Fig 1 presents a flowchart illustrating panel recruitment, number of panellists, age, sex, final pass-mark per exam, and item difficulty analysis for each examination, contributing to the consistency analyses of the inter-panel and panel/item difficulty.

## Panel recruitment

It has been debated which kind of subject matter experts to use in Angoff panels [21, 22]. In this study we identified a subject matter expert as a dentist educated around 10 years ago with a mix of current clinical experience and a background of teaching at the programme of dentistry, and the reference panellists were recruited consecutively before each examination from the Department of Dental Medicine at Karolinska Institutet during the period 2019-08-01 to 2020-08-19. The panels comprised clinical teachers in the dentistry programme, with the number of panellists in each test ranging from 10 to 12, while the subject matter experts were a mix of recurring and novice Angoff raters across the three examinations included in the study.

The external panel was recruited from the Faculty of Odontology at Malmö University, Sweden. Recruitment targeted clinical teachers in the dentistry programme and took place between 2021-05-15 and 2021-12-31 following the completion of the three theoretical examinations. The participants were given written information about the study in addition to an online information session. A total of ten subject matter experts were recruited, and those panellists who accepted the invitation were given both written instructions and all test items to review. Notably, external panel members did not have access to the reference panel ratings, and all were novices to the Angoff methodology.

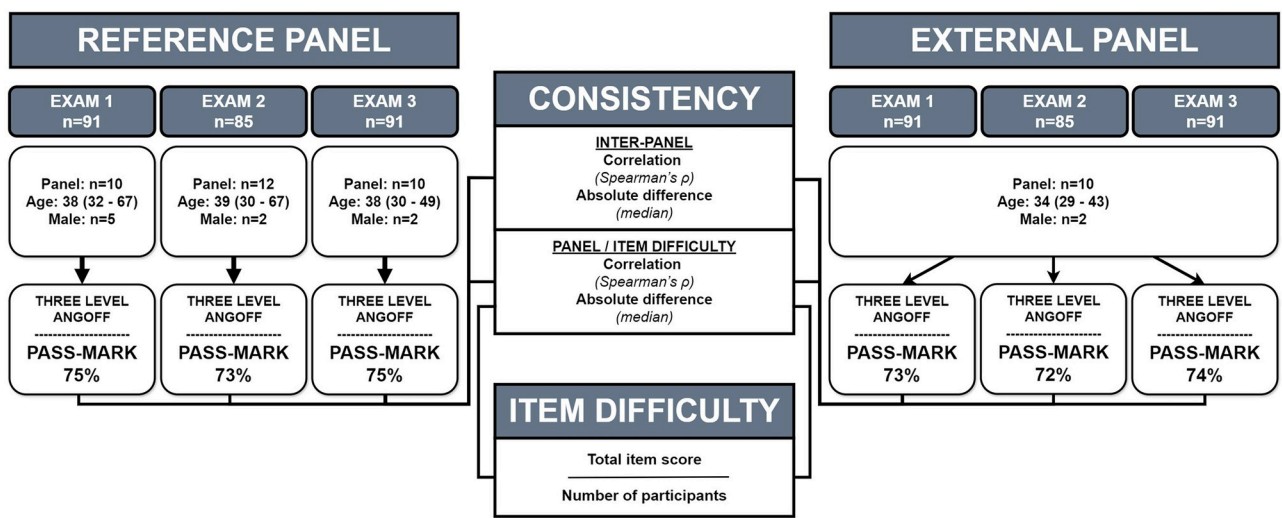

**Fig 1. Study design flowchart.** Flowchart illustrating panel recruitment, number of panellists, age, sex, final pass-mark per exam, and item difficulty analysis for each examination, contributing to the analysis of inter-panel and panel / item difficulty consistency.

### The simplified three-level Angoff item rating process

Test items in the theoretical test are primarily single best answer questions combined with very short essay and drag and drop question types. The panellists individually rated each question 'YES"NO' or 'MAYBE'. There was no training- or panel meetings for consensus. The three exams included between 85 and 91 questions.

### Variables

The theoretical proficiency test is a criterion-based examination. To systematically ensure adequate sampling of the professional qualifications, test items are created from a blueprint including the intended learning outcomes in the Swedish Degree of Master of Science in Dental Surgery [3]. These learning outcomes, here termed professional qualifications, establish a standard for the profession and constitute the foundation for the competency expected from minimally qualified dentists passing the exam. The blueprint from which the test items were created also includes the representation of dental disciplines, determined by the dimensioning within a Swedish syllabus of the study programme in dentistry. The alignment of professional qualifications and dental disciplines between and within the re-certification examinations has been presented in a previous publication [1]. The analyses in this paper cover a total of ten professional qualifications [3] and eight dental disciplines, namely *cariology*, *oral radiology*, *prosthodontics*, *periodontology*, *endodontology*, *oral medicine*, and *paediatric dentistry*. The dental disciplines presented in this study were selected from the major dental disciplines in the syllabus, as minor dental disciplines (2–3 questions on each exam) did notenable adequate analyses. The variable presented as *Angoff rating* is the calculated mean rating of each question from each panel. The Angoff ratings per professional qualification and dental discipline are the mean rating calculated from each question from the respective qualification or discipline, where the *item difficulty* is the calculated mean scores of the participants, whereas the item difficulty *per* professional qualification and dental discipline is the mean item difficulty calculated from the respective qualification or discipline.

The three options in the three-level Angoff are assigned weights as follows: 'YES' is weighted as 1, 'NO' is weighted as 0, and 'MAYBE' is weighted as .3. Total scores were calculated by summing the number of 'YES,' 'NO' and 'MAYBE' answers multiplied by their respective weights.

### Statistical analysis

Descriptive statistics were used to describe the number of items in each exam and demographic variables of the two panels. Nonparametric methods like medians, Spearman's rho, Mann-Whitney's test, and ordinal regression were used throughout since the method of calculating total scores implied that the normal distribution would be a poor model. The consistency of the two panel item ratings was analysed by Spearman's correlation coefficient ($\rho$). Spearman's $\rho$ was also used to analyse the consistency between the panel ratings and the participant group scores. The consistency between panel ratings, and the panel's rating versus participant group score, was analysed within the total items in each exam, both for dental discipline and professional qualification. A p-value $\leq 0.05$ was regarded as statistically significant. To explore differences between panel ratings, i.e., which panel rated the highest score, the absolute difference was calculated from the median of each panel's final rating. To explore differences between panel ratings and the participant group score, i.e., if the panels overestimated or underestimated the performance of participants, the absolute difference was calculated from the median of each panel's final rating and the participants' group score. Furthermore, the significance of median differences were determined using Mann-Whitney's test, and regression

models were used to visualise the relationship between the panel ratings and participant scores. Ordinal regression of the panellists' responses ('YES"NO' or 'MAYBE') was moreover used to explore the impact of age and sex of the subject matter experts, where age was categorised and analysed in both groups (over or under the median age of 36 years) and three groups (29–33 years, 34–39 years, and 40–67 years). All analyses were conducted using R version 4.3.2 (R Foundation for Statistical Computing, Vienna, Austria).

### Ethical considerations

Ethical approval for this study was granted by the Swedish Ethical Review Authority (Protocol identification number: 2019–06028, April 2[nd], 2020). Data collected from the theoretical examination (scores) and reference panellists were gathered as part of regular examination, and consent was not collected, as the collected data do not contain sensitive information and were delivered anonymised for data curation and analysis. Informed consent was obtained from all participants in the external Angoff panel, as their ratings are not part of regular examination procedure. Finally, rater data from the external panel were anonymised upon collection, and a random ID was provided to all external panellists for rating submissions. The study is reported according to the Strengthening the Reporting of Observational Studies in Epidemiology (STROBE) statement guidelines [23].

## Results

In Fig 1 the number of items, number of participating subject matter experts, age, sex, and final pass mark per test and panel are presented. The two panels consisted primarily of women, which reflects the current sex-distribution of dental teachers in both universities. The age of the reference panel was slightly higher compared to the external panel. Analyses of the potential effects of age and sex showed no statistically significant difference between groups.

### Inter-panel consistency

The ratings of the two panels of experts from different universities evaluated the difficulty of the items similarly across all exams, dental subjects, and professional qualifications ($\rho \geq 0.70$, p < .001). Although the reference panel set slightly higher pass marks overall (73% to 75%) compared to the external panel (72% to 74%), the difference in how the two panels rated individual exam questions was not significant. This consistency suggests that the standards for judging the exams were aligned between the two panels, despite being from different institutions. The average item correlation, the correlation *per* dental discipline and professional qualification, and the absolute difference between panels by exam are presented in Table 1.

### Consistency of panel rating and item difficulty

There was a strong relationship between how the panels rated the exam items and how the participants performed in the dental disciplines of Cariology, Endodontology, Oral Radiology, Periodontology, and Paediatric Dentistry ($\rho$ 0.40–0.69, $p \leq 0.001$). Prosthodontics showed the weakest correlation between panel ratings and participant scores(Reference = 0.148, External = 0.059). Orofacial Medicine had a moderate correlation(Reference = 0.306, External = 0.399), with the external panel rating slightly lower than participant performance (External = -0.022).Table 2 shows the correlation between the Angoff ratings and the item difficulty within each exam and dental discipline.

**Table 1. Average item correlations and absolute item differences between reference and external panel by exam, divided into dental disciplines, professional qualifications, and total items (n = 267).**

| Variable | Inter-panel item correlation | | | | | | | | Inter-panel item difference | | | | | | | |
|---|---|---|---|---|---|---|---|---|---|---|---|---|---|---|---|---|
| | Exam 1 | | Exam 2 | | Exam 3 | | All items | | Exam 1 | | Exam 2 | | Exam 3 | | All items | |
| | *(n = 91)* | | *(n = 85)* | | *(n = 91)* | | *(n = 267)* | | *(n = 91)* | | *(n = 85)* | | *(n = 91)* | | *(n = 267)* | |
| | *p* | *p*-value | *p* | *p*-value | *p* | *p*-value | *p* | *p*-value | *Dif.* | *p*-value | *Dif.* | *p*-value | *Dif.* | *p*-value | *Dif.* | *p*-value |
| **Dental discipline** | | | | | | | | | | | | | | | | |
| Cariology | **.96** | **< .001** | **.79** | **< .001** | **.81** | **< .001** | **.87** | **< .001** | .028 | 1 | .050 | .254 | -.010 | .940 | .038 | .603 |
| Endodontology | **.92** | **< .001** | N/A | | **.86** | **< .001** | **.94** | **< .001** | .053 | .596 | -.027 | .789 | -.100 | .466 | .001 | .485 |
| Oral radiology | **.87** | **< .001** | **.93** | **< .001** | **.79** | **< .001** | **.90** | **< .001** | .063 | .247 | .010 | .742 | -.047 | .271 | .015 | 1 |
| Orofacial medicine | **.78** | **< .001** | **.79** | **< .001** | **.81** | **< .001** | **.80** | **< .001** | .039 | .631 | .080 | .176 | .074 | .364 | .011 | .574 |
| Periodontology | **.81** | **< .001** | **.87** | **< .001** | **.87** | **< .001** | **.87** | **< .001** | .041 | .344 | .077 | .448 | -.038 | .570 | -.013 | .852 |
| Paediatric | **.91** | **< .001** | **.93** | **< .001** | **.94** | **< .001** | **.93** | **< .001** | -.021 | .406 | -.048 | 1 | -.017 | .759 | .011 | .916 |
| Dentistry | | | | | | | | | | | | | | | | |
| Prosthodontics | **.83** | **< .001** | **.88** | **< .001** | **.83** | **< .001** | **.88** | **< .001** | .031 | .732 | -.026 | .947 | .100 | .172 | .077 | .410 |
| **Professional qualification** | | | | | | | | | | | | | | | | |
| PQ 1 | **.90** | **< .001** | **.89** | **< .001** | **.84** | **< .001** | **.88** | **< .001** | .030 | .912 | .058 | .771 | .003 | .912 | .040 | .519 |
| PQ 2 | **.91** | **< .001** | **.83** | **< .001** | **.69** | **< .001** | **.88** | **< .001** | .015 | .971 | .013 | .539 | .017 | .821 | .031 | .983 |
| PQ 3 | **.89** | **< .001** | **.89** | **< .001** | **.88** | **< .001** | **.89** | **< .001** | .050 | .850 | -.011 | .821 | .024 | .879 | .057 | .603 |
| PQ 6 | **.87** | **< .001** | **.91** | **< .001** | **.86** | **< .001** | **.88** | **< .001** | .101 | .353 | .047 | .582 | -.007 | .481 | .041 | .466 |
| PQ 7 | N/A | | **.73** | **< .001** | **.90** | **< .001** | **.89** | **< .001** | -.040 | .820 | .007 | .842 | -.067 | .446 | -.048 | .568 |
| PQ 8 | N/A | | N/A | | N/A | | **.90** | **< .001** | .000 | .399 | -.100 | .409 | .300 | .148 | .034 | .767 |
| PQ 9 | **.93** | **< .001** | **.93** | **< .001** | N/A | | **.89** | **< .001** | .071 | .520 | -.028 | .146 | .033 | .850 | .004 | .627 |
| PQ 11 | **.87** | **< .001** | **.84** | **< .001** | **.83** | **< .001** | **.88** | **< .001** | .105 | .280 | .015 | 1. | -.029 | .650 | .032 | .633 |
| PQ 13 | **.89** | **< .001** | **.91** | **< .001** | **.86** | **< .001** | **.88** | **< .001** | -.006 | .631 | -.008 | .771 | -.031 | .597 | -.004 | .949 |
| PQ 14 | **.81** | **< .001** | **.86** | **< .001** | **.92** | **< .001** | **.91** | **< .001** | -.006 | .880 | -.049 | .895 | -.097 | .677 | -.034 | .663 |
| **Total** | **.89** | **< .001** | **.89** | **< .001** | **.83** | **< .001** | **0.88** | **< .001** | .057 | .912 | 0.064 | .771 | -.002 | .912 | .032 | .755 |

**Statistically significant results in bold**. Inter-panel correlations (Spearman rho (p)): ≥0.70 Very strong relationship, 0.40–0.69 Strong relationship, 0.30–0.39 Moderate relationship, 0.20–0.29 Weak relationship, 0.01–0.19 No or negligible relationship. Dif: Inter-panel absolute item difference (median absolute difference). Positive absolute difference indicates a higher Angoff score for the reference panel and negative absolute item difference indicates a higher Angoff score for the external panel. N/A = Professional qualifications with less than eight items were excluded. A full list of professional qualifications explored can be found in the supporting information.

## Discussion

The study investigated the three-level Angoff item ratings of two independent panels across three theoretical tests. The results showed strong to very strong item correlation with no significant absolute difference between the two panels'item ratings of the three exams, item ratings within the dental disciplines or item ratings within the professional qualifications. As dental schools in Sweden independently establish their syllabi, the results were not certain beforehand. The results strongly suggest that even with the panels being recruited from independent dental schools, there is alignment in the expectations regarding the standards of a minimally qualified yet still acceptable dentist. Thus, the null hypothesis was not contradicted.

The theoretical examinations are a standard set concerning what is expected from a Swedish dentist, i.e., the professional qualifications and curricula for a Swedish degree in dental surgery [1–3]. In some specific dental disciplines, e.g., Prosthodontics and Orofacial Medicine, the relationship between the Angoff ratings and the item difficulty was moderate to negligible. Generally, the participants' median score did not reach the Angoff median set for either of the two panels. The results are in line with our previous exploration of the theoretical

**Table 2. Average item correlations and absolute item differences between panels and item difficulty in items per dental discipline and total items (n = 267).**

| Variable | Average Item correlation | | | | Absolute Item Difference | | | |
|---|---|---|---|---|---|---|---|---|
| | Reference panel (*n = 267*) | | External panel (*n = 267*) | | Reference panel (*n = 267*) | | External panel (*n = 267*) | |
| | ρ | *p*-value | ρ | *p*-value | *Dif.* | *p*-value | *Dif.* | *p*-value |
| Dental discipline | | | | | | | | |
| Cariology | **.623** | **< .001** | **.555** | **< .001** | **.091** | **.001** | **.121** | **.002** |
| Endodontology | **.709** | **< .001** | **.584** | **.001** | **.211** | **< .001** | **.171** | **< .001** |
| Oral radiology | **.471** | **< .001** | **.714** | **< .001** | **.218** | **< .001** | **.172** | **< .001** |
| Orofacial medicine | .306 | .074 | **.399** | **.018** | .114 | .15 | -.022 | .991 |
| Periodontology | **.640** | **< .001** | **.783** | **< .001** | .127 | .077 | .120 | .144 |
| Paediatric Dentistry | **.614** | **< .001** | **.601** | **< .001** | **.153** | **< .001** | **.208** | **< .001** |
| Prosthodontics | .148 | .428 | .059 | .752 | **.108** | **.027** | .031 | .231 |
| **Total** | **.493** | **< .001** | **.448** | **< .001** | **.108** | **< .001** | **.120** | **< .001** |

**Statistically significant results in bold**. Inter-panel correlations (Spearman rho (p)): ≥0.70 Very strong relationship, 0.40–0.69 Strong relationship, 0.30–0.39 Moderate relationship, 0.20–0.29 Weak relationship, 0.01–0.19 No or negligible relationship. Dif: Inter-panel absolute item difference (median absolute difference). Positive absolute difference indicates a higher Angoff score for the reference panel and negative absolute item difference indicates a higher Angoff score for the external panel. N/A = Professional qualifications with less than eight items were excluded. A full list of professional qualifications explored can be found in the supporting information.

examinations, i.e., participants who pass the theoretical examination still can show knowledge gaps in single dental disciplines [1]. Dental education and licensure processes vary across countries, and efforts are made to harmonise dental education in the EU and worldwide [24–27]. Consequently, certain items, in a Swedish context, may be anticipated by panellists as answered correctly by the minimally qualified test taker, whilst in other educational systems these items may not be part of the standard syllabus or are regarded as specialist competence. It is important to emphasise that the purpose of the theoretical examinations is to differentiate participants that have the knowledge and skills that meet the standards of a Swedish degree in Dental Surgery.

In the present study 10–12 panellists were used, a decision partly grounded in evidence-based practice since the recommended panel size typically falls within the range of five to fifteen members [21, 28]. This choice was also influenced by practical considerations, given the challenges associated with recruiting enough subject matter experts from a single institution for each theoretical examination. Whilst previous studies have suggested that panellist discussions can reduce variability and improve intra-rater agreement by presenting test data [29, 30], such measures would require substantial resources, making them impractical for regular examinations. The approach used in this study allowed for the inclusion of new item raters, easily integrated into the process with written instructions alone. The recruitment and standard-setting strategies used in our study could effectively expand the pool of available subject matter experts without necessitating the extensive resources typically associated with the original Angoff method and its modifications. Our findings align with earlier research indicating that individual ratings without consensus save time [31]. However, it is important to acknowledge the trade-offs involved in skipping consensus meetings. The method may not alter pass rates compared to consensus methods [31], but it limits opportunities for in-depth discussions on item relevance and difficulty. Moreover, the absence of consensus meetings may reduce the support available to new panellists, potentially affecting confidence in raters and the consistency of ratings. To address these concerns, future studies could explore the development of supplementary training materials or online platforms that allow for some level of interaction

without the full resource demands of traditional consensus meetings. Involving external panellists, from all national universities with a dental programme, is therefore recommended to mitigate the potential influence of specific practices or biases from individual institutions. This approach may also enhance the credibility of the process by ensuring that the expectations for the minimally qualified but acceptable dentist are consistent at a national level, thereby maintaining accuracy of assessments in evaluating the necessary competencies for dental practice. Whilst the modified approach to recruitment and standard setting in this study offers a practical solution to resource constraints, it is essential to consider the potential impacts on the quality of the standard-setting process. Balancing the need for efficiency with the importance of examination validity will be crucial in further refining this method.

## Limitations

The results suggest that even with the panels being recruited from independent dental schools, there is alignment in the expectations regarding the standards of a minimally qualified yet still acceptable dentist. However, the analysis of consistency is one of many ways to explore how the two panels rated the test items. It is important to clarify that the analyses of panellists' ratings do not explore if the panellists agree with each other, as agreement was not the primary outcome of this study. Our main purpose was to compare the two panels final ratings, and even though we did not explore agreement, the average results indicate that the panellists' combined ratings ordered the items similarly. Furthermore, the average absolute item differences between panels also indicate that panellists rated items similarly.

A weakness of the study is that item relevance is not explored. The Angoff ratings do not provide specific insight into item relevance, i.e., an item could be rated as "YES" but still be irrelevant for evaluating the competence of a dental practitioner. One way to further explore item relevance is to have the subject matter expert also rate the relevance of the items, while another method suggested for rating relevance, alongside difficulty, is the Ebel approach [8].

This study indicated consistency between panels rating the same exams using the three-level Angoff, and a high inter-rater reliability in the reference panel was also shown in a previous study [1]. The findings suggest that the three-level Angoff method can be used effectively in contexts where time and resources are limited. However, there are concerns about reduced variability when using only three levels in rating, where the original Angoff, which uses 1–100 for ratings, may yield a more reliable outcome [17]. How panel members conceptualise a minimally qualified candidate may also be affected by experience and training in the Angoff methodology [15, 32]. The panellists were instructed to regard whether the least qualified but still acceptable candidate will answer the item correctly, potentially offering panellists a clearer and more understandable scenario for item rating [33], especially for new raters. To ensure that the compromise inherent to the three-level Angoff is reliable, it is important to evaluate the effect of consensus meetings held to discuss deviant items (where panellists disagree to a larger extent) and help introduce new panellists to the method. Thus, additional studies are needed to fully understand the pros and the cons of the three-level Angoff method.

To further investigate the validity of item ratings from a national perspective, we plan to invite other Swedish study programmes in dentistry to participate in the regular Angoff rating before upcoming theoretical exams. These ratings will be analysed as part of quality assurance.

## Conclusion

The expectations of the minimally qualified but still acceptable dentist were comparable between the two independent panels across the three theoretical examinations here explored. The alignment between the panels indicates valid, reliable standards across institutions, despite

the independent syllabi of the two study programmes. However, while there is an alignment, differences in ratings remain. Consequently, involving multiple institutions in future standard-setting processes could help ensure that the standards reflect a broader range of educational practices, supporting credibility of licensure examinations.

## Supporting information

**S1 Checklist.**
(PDF)

## Acknowledgments

We thank the candidates who took part in the tests and the Angoff panellists. Ida Hed Myrberg and Henrike Häbel contributed to the process of data curation.

## Author Contributions

**Conceptualization:** Jesper Dalum, Liselotte Paulsson, Nikolaos Christidis, Klas Karlgren, Charlotte Leanderson, Gunilla Sandborgh-Englund.

**Data curation:** Jesper Dalum, Mikael Andersson Franko.

**Formal analysis:** Jesper Dalum, Mikael Andersson Franko, Gunilla Sandborgh-Englund.

**Funding acquisition:** Gunilla Sandborgh-Englund.

**Investigation:** Jesper Dalum.

**Methodology:** Jesper Dalum, Liselotte Paulsson, Nikolaos Christidis, Mikael Andersson Franko, Klas Karlgren, Charlotte Leanderson, Gunilla Sandborgh-Englund.

**Project administration:** Jesper Dalum, Liselotte Paulsson, Gunilla Sandborgh-Englund.

**Resources:** Gunilla Sandborgh-Englund.

**Software:** Jesper Dalum.

**Supervision:** Nikolaos Christidis, Klas Karlgren, Charlotte Leanderson, Gunilla Sandborgh-Englund.

**Visualization:** Jesper Dalum, Liselotte Paulsson, Mikael Andersson Franko.

**Writing – original draft:** Jesper Dalum.

**Writing – review & editing:** Jesper Dalum, Liselotte Paulsson, Nikolaos Christidis, Mikael Andersson Franko, Klas Karlgren, Charlotte Leanderson, Gunilla Sandborgh-Englund.

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
