## [Decision Letter · Decision Letter 0]

1 Aug 2024

PONE-D-24-15178Consistency between inter-institutional panels using a three-level Angoff - standard setting in licensure tests of foreign-trained dentists in SwedenPLOS ONE

Dear Dr. Dalum,

Thank you for submitting your manuscript to PLOS ONE. After careful consideration, we feel that it has merit but does not fully meet PLOS ONE’s publication criteria as it currently stands. Therefore, we invite you to submit a revised version of the manuscript that addresses the points raised during the review process. Please note that one of our external Referees has recommended rejection of your submitted draft ("The reasons for that (rejection) are manifold and could be obtained from the comments. A lack of readability and clarity are the first to be mentioned. Moreover the scientific impact of this study is very low."), while our second Reviewer obviously seem satisfied (at least to some major extent). Therefore, I have double-checked your paper (see comments forwarded as R #1). You surely will agree that with reference to the numerous shortcomings having been identified with your draft, your submitted manuscript is not considered ready to proceed. Please take the opportunity to carefully revise your paper, and stick to each and every comment, to speed up this review process. Remember that Plos One's policy is to go for one round of revisions only, to avoid any re-re-review, so ensure re-submitting a high-quality re-submission. 

We look forward to receiving your revised manuscript.

Kind regards,

Andrej M Kielbassa

Academic Editor

PLOS ONE

Journal Requirements:

2. Please note that in order to use the direct billing option the corresponding author must be affiliated with the chosen institute. Please either amend your manuscript to change the affiliation or corresponding author, or email us at plosone@plos.org with a request to remove this option.

Reviewers' comments:

Reviewer's Responses to Questions

**Comments to the Author**

1. Is the manuscript technically sound, and do the data support the conclusions?

Reviewer #1: Partly

Reviewer #2: Yes

Reviewer #3: Partly

2. Has the statistical analysis been performed appropriately and rigorously? 

Reviewer #1: Yes

Reviewer #2: Yes

Reviewer #3: Yes

3. Have the authors made all data underlying the findings in their manuscript fully available?

Reviewer #1: No

Reviewer #2: Yes

Reviewer #3: Yes

4. Is the manuscript presented in an intelligible fashion and written in standard English?

Reviewer #1: Yes

Reviewer #2: Yes

Reviewer #3: No

5. Review Comments to the Author

Reviewer #1: Abstract

- Please revise, and stick to Journal style.

- No subheadings mandatory. Please consult some recently published Plos One papers.

- Please provide exact P values here. Remember that terms like "very strong relationship" and "scores were overall lower" would seem too vague, and do not provide sound information.

- With your revision, please stick to the allowed word maximum, to provide as much information as possible.

Intro

- "To address this, the present study aims to investigate (...)." This study has been finished hasn't it? Please switch to past tense.

- No subheadings with this section. -

- Please elaborate bot aims and objectives with this section. Do not provide a literature review here.

- A sound and valid null hypothesis is missing at the end of this section.

Meths

- "(...) by two separate panels; The reference panel was (...)." Please revise for typos.

- Details of statistical program missing.

Results

- With you full text, please provide exact results. See "weakest relationship" or "moderate relationship" - you surely will agree that this would seem hard to understand.

Disc

- Stick to H0 when starting this section.

- This section would seem perfectible. There would be many more aspects to discuss. Please revise carefully.

Concl

- With your revision, please exclusively stick to your aims.

Refs

- This list must be adopted to Journal style.

In total, this submission would seem interesting, is considered easily intelligible, and should be worth following after revision, depending on the latter's quality. Please stick to the comments provided above, and revise carefully.

Reviewer #2: Title page:

- It would be helpful to potential readers if the title of the publication already indicated the type of study that was involved in the publication.

- Please note the formal requirements of PLoS One regarding the font size and font type of your headings, please see "Abstract".

- Please use either American or British English in your manuscript, but do not mix the two.

Introduction:

- "To address this, the present study aims to investigate the consistency of three-level Angoff ratings of the proficiency test between two independent panels of raters from two universities." - "To address this, the aim of the present study was to investigate..."

- Please provide a sound and valid null hypothesis at the end of your Introduction section, which should be discussed at the beginning of the Discussion section.

Materials and methods:

- With the material used (i.e., the statistical software), please use general names with your text, followed by (brand name; manufacturer, city, ST[ate - abbreviated, if US], country) in parentheses. Stick to semicolon.

- Which test did you use to determine the (missing) normal distribution?

- The significance level α is missing. I guess it was 0.05?

- The ethics committee vote date is missing.

Discussion:

- The mentioning of statements regarding the null hypothesis at the beginning of this section is missing. Remember that the null hypothesis can be rejected or not rejected.

References:

- Please follow the author guidelines for ALL your References and provide all necessary data such as the correct abbreviated journal name, volume, issue, page numbers etc. Please carefully revise your data.

- Please make sure the title of each reference you access online is complete, correct, and includes access date and link.

- Please provide the correct and complete names of authors of websites used, e.g. References 2-4.

Reviewer #3: The authors present valuable insight into the Swedish process of re-certifying dentists trained outside the EU. Two expert panels assessed three different types of examinations using a modified Angoff method for setting exam pass-marks. The results suggest an overall high correlation between both (expert) groups in the majority of items although the syllabus is variable from one university to another. It was concluded that simplification of the Angoff method is appropriate, less time consuming and resource-saving compared to the originally described procedure.

The paper is well organized and concise. However, information is provided in a cumbersome style, which is hard to read and confusing at times. To me the paper needs extensive linguistic revision especially in the Materials and Methods and Results sections. There is a lack of clarity, consistency and readability in particular.

• For the above mentioned reasons the abstract needs to be rewritten. The results are almost incomprehensible. Who are the participants in this study? I assume you refer to the external panel but this is nowhere mentioned and confuses the reader to the maximum.

Why do you consider the modified Angoff method to be “promising” compared to the original one when you have not even investigated both methods?

• In the Results section, the paragraph starting on page 6 l151-page 7 l158 is incomprehensible at all and probably needs visualisation. (For you who are dealing with these closely related technical terms on a daily basis this all might be self-explanatory which is obviously not the case for your readers).

• Description of the study’s limitations is by far the best paragraph in the paper which arises the question whether an overall “similarity” (since agreement was not assessed in the first place) between to different panels assessing pass-marks of three exams is worth to be reported in a high-standard international publication?

• The closing sentence again refers to the traditional Angoff method which again has not been investigated in this study.

6. PLOS authors have the option to publish the peer review history of their article (what does this mean?). If published, this will include your full peer review and any attached files.

Reviewer #1: No

Reviewer #2: No

Reviewer #3: **Yes: **PD Dr. Hans Juergen Jochen Michael Wicht

---

## [Author Response · Author response to Decision Letter 0]

9 Sep 2024

PONE-D-24-15178

Consistency between inter-institutional panels using a three-level Angoff - standard setting in licensure tests of foreign-trained dentists in Sweden

We would like to thank the editor and reviewers for taking the time to review the manuscript, and considering the manuscript for publication. We hope that the changes made to the manuscript is up to you expectations.

Responses to comments, in detail, can be found in the document Response to reviewers. Below is a summarised version of the comments to specific reviewer and editor comments.

Reviewer 1

We have revised the abstract to conform with the journal's style. We have used the recently published article in formatting our manuscript. The exact p-values are now provided in the abstract. 

The Introduction has been revised to reflect that the study is finished. We have removed the subheadings from the Introduction to align with the journal's style requirements. The null hypothesis has been explicitly stated at the end of the Introduction. Additionally, aims and objectives have been elaborated. 

We have revised the text to correct any typographical errors. We have included detailed information about the statistical software used in the analyses, including the version, manufacturer, and location. The exact results from statistical analyses have been provided throughout the text.

We have revised the Discussion section to focus on the null hypothesis and expanded the discussion to cover additional relevant aspects. 

The Conclusion has been revised to strictly adhere to the study's aims. 

References have been reformatted to comply with the journal's guidelines. 

Reviewer 2

Thank you for your suggestion. We have revised the title to reflect the nature of the study more clearly. We have reviewed and adjusted the font size and type of all headings throughout the manuscript to comply with PLoS One formatting guidelines. We have also revised the text to British English.

We have revised the text in the introduction. The null hypothesis has been explicitly stated at the end of the Introduction. Additionally, aims and objectives have been elaborated. We have revised the Discussion section to focus on the null hypothesis and expanded the discussion to cover additional relevant aspects.

We have revised the Material and Methods section. We now include the brand, manufacturer, and location details for the statistical software used, as suggested. No formal test of possible normal distribution was made. Instead, we concluded that the normal distribution would be a poor model based on the way total scores were calculated. This has been clarified in the manuscript. We have revised the text to include the significance level. The date of the ethics committee approval has been included in the manuscript under Ethical consideration.

The Discussion begins with a reference to the null hypothesis. See revisions in first section of Discussion.

All references have been revised to follow the journal’s style, including online access information where relevant

Reviewer 3

The abstract and Methods section has been rewritten for clarity.

We have clarified why the modified Angoff method is considered promising, despite not directly comparing it to the original method. See Discussion and limitations section. We have also revised the conclusions regarding the reference to the original Angoff method. 

We have revised the results section to provide clarity, adding an explanatory text in the beginning of each paragraph in the results section. 

The Discussion section has been revised to ensure the study’s contribution is clear, in the context of the study’s limitations.

The Conclusion has been revised to remove ambiguous references to the traditional Angoff method and focuses solely on the findings.

---

## [Decision Letter · Decision Letter 1]

15 Sep 2024

PONE-D-24-15178R1Consistency between inter-institutional panels using a three-level Angoff - standard setting in licensure tests of foreign-trained dentists in Sweden: A cross-sectional studyPLOS ONE

Dear Dr. Dalum,

Thank you for submitting your manuscript to PLOS ONE. After careful consideration, we feel that it has merit but does not fully meet PLOS ONE’s publication criteria as it currently stands. Therefore, we invite you to submit a revised version of the manuscript that addresses the points raised during the review process. Please note that one of the previous external reviewers rejecting your manuscript has declined to re-review, without double-checking your revised draft. The second reviewer (recommending revisions with the first round of reviews) has rejected your revised version, mainly due to formal aspects. No doubt, the latter have to be followed, and you will find the respective comments given below. Please note that our referees have raised several points that do jeopardize the outlook for publication of your manuscript, so please remember that these concerns must be fully addressed, and re-review of the manuscript will be necessary and extensive. Additionally, I have double checked your draft, and have commented on the revised version (for details, please see below). I am confident that you will stick to those aspects, and you will re-submitted an improved version of your manuscript clarifying the missing aspects. Please note that usually only one revision would seem acceptable, and repeated re-review is discouraged.

We look forward to receiving your revised manuscript.

Kind regards,

Andrej M Kielbassa

Academic Editor

PLOS ONE

Additional Editor Comments:

- Unfortunately, the Co-Authors have not provided a point-by-point response to the previous Reviewers’ comments. Please go back to the recent recommendations, and clarify how you have responded to the recently forwarded recommendations.

- „Angoff - standard“ – please double check for correct spacebar use.

- „The reference panel ratings originated from the panel used to consecutively set the pass mark for regular examinations. The external panel was recruited, as part of the study, from a dental school at a university in another region in Sweden after the examinations.“ Please note that this would seem hard to understand; it would seem that you have compared regular students from two different universities. In particular when reflecting on your thoughts given with your Abstract section („This study aimed to determine the consistency of ratings by independent panels at two Swedish universities in re-certification exams for non-EU trained dentists“), this should be clarified. Please clearly define your reference and external groups.

- The Co-Authors state that „A cross-sectional design was used to analyze rating consistency of two separate panels.“ Please note that a cross-sectional study design is a type of observational study design. First, this would call for pre-registration (as with all observational studies, see https://www.ncbi.nlm.nih.gov/pmc/articles/PMC2952011/ ). Second, cross sectional studies should be reported according to STROBE (see https://www.equator-network.org/wp-content/uploads/2015/10/STROBE_checklist_v4_cross-sectional.pdf), along with a dedicated information given with your manuscript. You surely will agree that both aspects would seem missing with your revised and re-submitted draft.

- „The panellists in the reference panel were recruited consecutively before each examination during 2019-08-01 to 2020-08-19 (...).“ With reference to the thoughts given above, this would be a group of people who share a common characteristic or experience within a defined period (e.g., have been examined. This usually would be a cohort. Consequently, this would be a cohort study comparing two subgroup cohorts (reference and external). Please double check, and give reasons why you have called this study a cross sectional study.

- „The two panels consisted primarily of women.“ Please provide reasons.

- With your Conclusions, please stick exclusively to your aims (these were „This study aimed to determine the consistency of ratings by independent panels (...)“, or „This study aimed to determine whether the expectations of minimally qualified dentists are aligned, as reflected by the ratings from two universities using the three-level Angoff method“). Do not simply repeat your results here. Do not speculate. Instead, provide a reasonable and generalizable extension of your outcome.

- „Additional studies are needed to fully understand the pros and the cons of the three-level Angoff method.“ This is not considered a conclusion from your study. Please, these aspects might be discussed with your Discussion section; please delete these thoughts from your conclusions.

- References still have not been adapted to Journal style (please see previous comments, and compare your responses: „References have been reformatted to comply with the journal's guidelines“, and „All references have been revised to follow the journal’s style“). With your revisions, you surely want to make up this leeway. Please revise for English titles, and provide doi numbers.

Reviewers' comments:

Reviewer's Responses to Questions

**Comments to the Author**

1. If the authors have adequately addressed your comments raised in a previous round of review and you feel that this manuscript is now acceptable for publication, you may indicate that here to bypass the “Comments to the Author” section, enter your conflict of interest statement in the “Confidential to Editor” section, and submit your "Accept" recommendation.

Reviewer #1: All comments have been addressed

Reviewer #2: (No Response)

2. Is the manuscript technically sound, and do the data support the conclusions?

Reviewer #1: Yes

Reviewer #2: Yes

3. Has the statistical analysis been performed appropriately and rigorously? 

Reviewer #1: Yes

Reviewer #2: Yes

4. Have the authors made all data underlying the findings in their manuscript fully available?

Reviewer #1: Yes

Reviewer #2: Yes

5. Is the manuscript presented in an intelligible fashion and written in standard English?

Reviewer #1: Yes

Reviewer #2: No

6. Review Comments to the Author

Reviewer #1: The Co-Authors have satisfyingly revised their previous draft. Unfortunately, doi numbers still would seem missing with most of the references, but this should be perfectible in cooperation with the typesetter.

Reviewer #2: Dear Authors,

Abstract:

- No subheadings are required in the Abstract. Please note the comment from Reviewer #1 in the last review.

https://journals.plos.org/plosone/article?id=10.1371/journal.pone.0310004

https://journals.plos.org/plosone/article?id=10.1371/journal.pone.0228249

- Please put a space before and after a mathematical operator (e.g., minus, plus, greater than, less than) in the whole manuscript as well as before and after the hyphen.

- Please note the comment from Reviewer #1 regarding the mention of exact p values.

- Shouldn't the order of the p and Spearman's Rho values be the other way around (significance/correlation) according to your explanation? - "There was a statistically significant and strong correlation (ρ ≥ 0.70, p ≤ 0.05) between ..." - "There was a statistically significant and strong correlation (p ≤ 0.05, ρ ≥ 0.70) between..."

- Please report exact p values (e.g., p = .015, p < .XYZ), unless p is < .001. In accordance with the APA style, p values smaller than p < .001 are written as p < .001 (https://www.graphpad.com/support/faq/how-to-report-p-values-in-journals).

- "There was a statistically significant and strong correlation (ρ ≥ 0.70, p ≤ 0.05) between the ratings of the two

independent panels, with no significant differences ..." - Here, the exact p value is missing as well.

- Please re-check your punctuation skills in English.

Introduction:

- "The null hypothesis was that there will be no significant difference..." - "The null hypothesis was that there would be no significant difference..."

Materials and methods

- Please put a space before and after a mathematical operator (e.g., minus, plus, greater than, less than) in the whole manuscript as well as before and after the hyphen.

- Please consistently omit the zero before the decimal point for the p values and the correlation coefficient because the statistic cannot be greater than 1 (APA style), and please do not change the notation within the manuscript.

Results

- Please put a space before and after a mathematical operator (e.g., minus, plus, greater than, less than) in the whole manuscript as well as before and after the hyphen.

- Please report exact p values (e.g., p = .015, p < .XYZ), unless p is < .001. In accordance with the APA style, p values smaller than p < .001 are written as p < .001 (https://www.graphpad.com/support/faq/how-to-report-p-values-in-journals).

- Please consistently omit the zero before the decimal point for the p values and the correlation coefficient because the statistic cannot be greater than 1 (APA style), and please do not change the notation within the manuscript.

Conclusion

This chapter is formulated in very general terms.

Refs

- Please revise this part with regard to up-to-dateness. For example, complete information is available on PubMed for Reference 1. https://pubmed.ncbi.nlm.nih.gov/36039793

I assume that it is the task of the authors to ensure that a revised version is up-to-date and free of errors.

7. PLOS authors have the option to publish the peer review history of their article (what does this mean?). If published, this will include your full peer review and any attached files.

Reviewer #1: No

Reviewer #2: No

---

## [Author Response · Author response to Decision Letter 1]

18 Oct 2024

PONE-D-24-15178R1

Consistency between inter-institutional panels using a three-level Angoff - standard setting in licensure tests of foreign-trained dentists in Sweden: A cross-sectional study

Revised title: Consistency between inter-institutional panels using a three-level Angoff-standard setting in licensure tests of foreign-trained dentists in Sweden: A cohort study

PLOS ONE

Dear Editor and reviewers,

Thank you for taking the time to review and to improve upon our manuscript. In the following text you will find our point-by-point response to the comments. The comments are also uploaded as a word document. We have also included an updated point-by-point response for the 1st revision in attached files.

The manuscript has been formatted according to reference article: https://journals.plos.org/plosone/article?id=10.1371/journal.pone.0228249. The manuscript has also been sent for language check at a third party.

Additional Editor Comments:

- Unfortunately, the Co-Authors have not provided a point-by-point response to the previous Reviewers’ comments. Please go back to the recent recommendations, and clarify how you have responded to the recently forwarded recommendations.

Reply: Point-by-point response to the first review is provided in the uploaded document PONE Response to reviewers 1st revision - point-by-point.

- „Angoff - standard“ – please double check for correct spacebar use.

Reply: Thank you, this is corrected. 

- „The reference panel ratings originated from the panel used to consecutively set the pass mark for regular examinations. The external panel was recruited, as part of the study, from a dental school at a university in another region in Sweden after the examinations.“ Please note that this would seem hard to understand; it would seem that you have compared regular students from two different universities. In particular when reflecting on your thoughts given with your Abstract section („This study aimed to determine the consistency of ratings by independent panels at two Swedish universities in re-certification exams for non-EU trained dentists“), this should be clarified. Please clearly define your reference and external groups.

Reply: The abstract has been rephrased to clarify the design of the study.

- The Co-Authors state that „A cross-sectional design was used to analyze rating consistency of two separate panels.“ Please note that a cross-sectional study design is a type of observational study design. First, this would call for pre-registration (as with all observational studies, see https://www.ncbi.nlm.nih.gov/pmc/articles/PMC2952011/ ). Second, cross sectional studies should be reported according to STROBE (see https://www.equator-network.org/wp-content/uploads/2015/10/STROBE_checklist_v4_cross-sectional.pdf), along with a dedicated information given with your manuscript. You surely will agree that both aspects would seem missing with your revised and re-submitted draft.

Reply: Yes, we agree. We acknowledge that cross-sectional studies are observational by nature and agree that pre-registration of medical studies and adherence to STROBE guidelines are important. This study was not pre-registered. A STROBE Statement has been added at the end of ethical consideration, and a document how the manuscript adheres to the STROBE guidelines is also added to the supplementary material. 

- „The panellists in the reference panel were recruited consecutively before each examination during 2019-08-01 to 2020-08-19 (...).“ With reference to the thoughts given above, this would be a group of people who share a common characteristic or experience within a defined period (e.g., have been examined. This usually would be a cohort. Consequently, this would be a cohort study comparing two subgroup cohorts (reference and external). Please double check, and give reasons why you have called this study a cross sectional study.

Reply: Thank you for pointing this out, we have adjusted the text accordingly in the title and in the first sentence of the methods section.

- „The two panels consisted primarily of women.“ Please provide reasons.

Reply: Thank you for your comment. The absolute majority (at least 75%) of the dental teachers are women in both dental schools involved. This is reflected by the sex distribution of the panels. We have added text “The two panels consisted primarily of women, which reflects the current sex-distribution of dental teachers in both universities.” In the beginning of the Results. Page 7, Row number 165-166.

- With your Conclusions, please stick exclusively to your aims (these were „This study aimed to determine the consistency of ratings by independent panels (...)“, or „This study aimed to determine whether the expectations of minimally qualified dentists are aligned, as reflected by the ratings from two universities using the three-level Angoff method“). Do not simply repeat your results here. Do not speculate. Instead, provide a reasonable and generalizable extension of your outcome.

Reply: Thank you, the Conclusions have been revised accordingly.

- „Additional studies are needed to fully understand the pros and the cons of the three-level Angoff method.“ This is not considered a conclusion from your study. Please, these aspects might be discussed with your Discussion section; please delete these thoughts from your conclusions.

Reply: Thank you for your comment, the statement has been removed from the conclusion section.

- References still have not been adapted to Journal style (please see previous comments, and compare your responses: „References have been reformatted to comply with the journal's guidelines“, and „All references have been revised to follow the journal’s style“). With your revisions, you surely want to make up this leeway. Please revise for English titles, and provide doi numbers.

Reply: Thank you for your comment, the references are now formatted correctly

Reviewer #1: The Co-Authors have satisfyingly revised their previous draft. Unfortunately, doi numbers still would seem missing with most of the references, but this should be perfectible in cooperation with the typesetter.

Reply: Thank you for your comment, the references are now formatted correctly

Reviewer #2: Dear Authors,

Abstract:

- No subheadings are required in the Abstract. Please note the comment from Reviewer #1 in the last review.

https://journals.plos.org/plosone/article?id=10.1371/journal.pone.0310004

https://journals.plos.org/plosone/article?id=10.1371/journal.pone.0228249

Reply: Thank you for your comment. Subheadings have been deleted in the abstract

- Please put a space before and after a mathematical operator (e.g., minus, plus, greater than, less than) in the whole manuscript as well as before and after the hyphen.

Reply: Thank you for your observation, this has been corrected in the text. 

- Please note the comment from Reviewer #1 regarding the mention of exact p values.

Reply: Yes, we have corrected this by adding exact p-values in the Tables. 

- Shouldn't the order of the p and Spearman's Rho values be the other way around (significance/correlation) according to your explanation? - "There was a statistically significant and strong correlation (ρ ≥ 0.70, p ≤ 0.05) between ..." - "There was a statistically significant and strong correlation (p ≤ 0.05, ρ ≥ 0.70) between..."

Reply: Thank you for noticing, we have corrected these sentences.

- Please report exact p values (e.g., p = .015, p < .XYZ), unless p is < .001. In accordance with the APA style, p values smaller than p < .001 are written as p < .001 (https://www.graphpad.com/support/faq/how-to-report-p-values-in-journals).

Reply Yes, we have corrected this by formatting the text and adding exact p-values in the Tables. 

- "There was a statistically significant and strong correlation (ρ ≥ 0.70, p ≤ 0.05) between the ratings of the two

independent panels, with no significant differences ..." - Here, the exact p value is missing as well.

Reply: Thank you for observing this. In running text, we refer to overall results. The p-values have been corrected. Exact p-values have been added in the revised Tables for reference.

- Please re-check your punctuation skills in English.

Reply: We have re-checked the punctuation, and hope this is OK now

Introduction:

- "The null hypothesis was that there will be no significant difference..." - "The null hypothesis was that there would be no significant difference..."

Reply Thank you, the text has been revised according to your comment.

Materials and methods

- Please put a space before and after a mathematical operator (e.g., minus, plus, greater than, less than) in the whole manuscript as well as before and after the hyphen.

Reply: We have checked the manuscript and have hopefully corrected these mistakes.

- Please consistently omit the zero before the decimal point for the p values and the correlation coefficient because the statistic cannot be greater than 1 (APA style), and please do not change the notation within the manuscript.

Reply: We have checked the manuscript and have hopefully corrected these mistakes.

Results

- Please put a space before and after a mathematical operator (e.g., minus, plus, greater than, less than) in the whole manuscript as well as before and after the hyphen.

Reply: We have checked the manuscript and have hopefully corrected these mistakes

- Please report exact p values (e.g., p = .015, p < .XYZ), unless p is < .001. In accordance with the APA style, p values smaller than p < .001 are written as p < .001 (https://www.graphpad.com/support/faq/how-to-report-p-values-in-journals).

Reply: We have checked the manuscript and have hopefully corrected these mistakes

- Please consistently omit the zero before the decimal point for the p values and the correlation coefficient because the statistic cannot be greater than 1 (APA style), and please do not change the notation within the manuscript.

Reply: We have checked the manuscript and have hopefully corrected these mistakes

Conclusion

This chapter is formulated in very general terms.

Reply: The conclusion section has been rewritten in order to provide a reasonable and generalizable extension.

Refs

- Please revise this part with regard to up-to-dateness. For example, complete information is available on PubMed for Reference 1. https://pubmed.ncbi.nlm.nih.gov/36039793

I assume that it is the task of the authors to ensure that a revised version is up-to-date and free of errors.

Reply: We’re sorry about these mistakes, and the reference list has been updated accordingly.

---

## [Decision Letter · Decision Letter 2]

25 Oct 2024

Consistency between inter-institutional panels using a three-level Angoff - standard setting in licensure tests of foreign-trained dentists in Sweden: A cohort study

PONE-D-24-15178R2

Dear Dr. Dalum,

We’re pleased to inform you that your manuscript has been judged scientifically suitable for publication and will be formally accepted for publication once it meets all outstanding technical requirements. Please not that one of our external reviewers has noted that still some minor issues would call for improvements; these should be perfectible together with the typesetter, so please pay special attention to the proofs.

Kind regards,

Prof. Dr. Dr. h. c. Andrej M Kielbassa

Academic Editor

PLOS ONE

Reviewers' comments:

Reviewer's Responses to Questions

**Comments to the Author**

1. If the authors have adequately addressed your comments raised in a previous round of review and you feel that this manuscript is now acceptable for publication, you may indicate that here to bypass the “Comments to the Author” section, enter your conflict of interest statement in the “Confidential to Editor” section, and submit your "Accept" recommendation.

Reviewer #1: All comments have been addressed

Reviewer #2: (No Response)

2. Is the manuscript technically sound, and do the data support the conclusions?

Reviewer #1: Yes

Reviewer #2: Yes

3. Has the statistical analysis been performed appropriately and rigorously? 

Reviewer #1: Yes

Reviewer #2: Yes

4. Have the authors made all data underlying the findings in their manuscript fully available?

Reviewer #1: Yes

Reviewer #2: Yes

5. Is the manuscript presented in an intelligible fashion and written in standard English?

Reviewer #1: Yes

Reviewer #2: Yes

6. Review Comments to the Author

Reviewer #1: With the help of the Reviewers, the Co-Authors have re-submitted a revised draft now, and the latter would seem ready to proceed.

Reviewer #2: Dear Authors,

your reference section unfortunately still contains errors and information that should not be listed, and unfortunately you have again not fully followed the Plos One author guidelines, i.e., “Epub” should not be listed and is even partially (according to PubMed) incorrect (see Ref. 17) and the page numbers should both be listed in full. It is your second revision.

7. PLOS authors have the option to publish the peer review history of their article (what does this mean?). If published, this will include your full peer review and any attached files.

Reviewer #1: No

Reviewer #2: No

---

## [Editor Report · Acceptance letter]

30 Oct 2024

PONE-D-24-15178R2 

PLOS ONE

Dear Dr. Dalum, 

I'm pleased to inform you that your manuscript has been deemed suitable for publication in PLOS ONE. Congratulations! Your manuscript is now being handed over to our production team.

Kind regards, 

on behalf of

Prof. Dr. med. dent. Dr. h. c. Andrej M Kielbassa 

Academic Editor

PLOS ONE